# Preventive Effects of Sustainable and Developmental Perioperative Oral Management Using the “Oral Triage” System on Postoperative Pneumonia after Cancer Surgery

**DOI:** 10.3390/ijerph18126296

**Published:** 2021-06-10

**Authors:** Hideki Sekiya, Yasuhiro Kurasawa, Kosuke Kaneko, Ken-ichiro Takahashi, Yutaka Maruoka, Yukihiro Michiwaki, Yoshimasa Takeda, Ryoichi Ochiai

**Affiliations:** 1Department of Oral Surgery, School of Medicine, Toho University, Tokyo 143-8541, Japan; kousuke.kaneko@med.toho-u.ac.jp (K.K.); ken514@med.toho-u.ac.jp (K.-i.T.); yukirom@musashino.jrc.or.jp (Y.M.); 2Department of Oral & Maxillofacial Surgery, Tokyo Medical & Dental University, Tokyo 113-0034, Japan; yasukura1408@gmail.com; 3Department of Oral & Maxillofacial Surgery, Center Hospital of the National Center for Global Health and Medicine, Tokyo 162-8655, Japan; ymaruoka@hosp.ncgm.go.jp; 4Department of Oral & Maxillofacial Surgery, Musashino Redcross Hospital, Tokyo 180-8610, Japan; 5Department of Anesthesiology (Omori), School of Medicine, Toho University, Tokyo 143-8541, Japan; yoshimasa.takeda@med.toho-u.ac.jp (Y.T.); roy.ochiai@gmail.com (R.O.)

**Keywords:** cancer surgery, oral care, oral triage, postoperative pneumonia, perioperative oral management, minimal human resources

## Abstract

Perioperative oral management is widely recognized in the healthcare system of Japan. Conventionally, the surgeon refers patients with oral problems to a dental or oral surgery clinic in the hospital. However, frequent in-house referrals were found to increase the number of incoming patients resulting in unsustainable situations due to an insufficient workforce. In 2011, the Center for Perioperative Medicine was established at our hospital to function as a management gateway for patients scheduled to undergo surgery under general anesthesia. The “oral triage” system, wherein a dental hygienist conducts an oral screening to select patients who need preoperative oral hygiene and functional management, was established in 2012. A total of 37,557 patients who underwent surgery at our hospital from April 2010 to March 2019 (two years before and seven years after introducing the system) were evaluated in this study. The sustainability and effectiveness of introducing the system were examined in 7715 cancer surgery patients. An oral management intervention rate of 20% and a significant decrease in the incidence of postoperative pneumonia (aOR = 0.50, *p* = 0.03) indicated that this system could be useful as a sustainable and developmental oral management strategy to manage surgical patients with minimal human resources.

## 1. Introduction

The development of minimally invasive surgical techniques has improved the chances of selecting an appropriate surgical treatment method, even in elderly patients with comorbidities. Therefore, the need for multidisciplinary care to conduct the procedures safely is gaining recognition. Since the establishment of a management gateway called the Center for Perioperative Medicine in April 2011, multidisciplinary approaches have been used to provide risk-free and reliable surgery to patients in our hospital. The perioperative oral management (oral care) section, which plays a significant role in these approaches, was implemented in April 2012.

Simultaneously, the “perioperative oral management fee” was listed in the fee schedule for insured medical treatments (universal health insurance system in Japan) in April 2012. Since then, various hospitals affiliated with medical schools and municipal hospitals initiated the perioperative management of oral hygiene and function at the time of surgery, under general anesthesia, through collaborations between the medical and dental teams.

In general, it is common for the surgeon in charge to refer patients to a dental or oral surgery clinic at the hospital (hereinafter referred to as the conventional method). However, this referral is made solely by a surgeon; therefore, in some instances, patients with poor oral hygiene may not visit a dental clinic or receive oral treatment before the surgery due to a last-minute referral. Moreover, as the in-house referral to dental professionals has become a commonly known approach, it is predicted that the referral will be conducted without evaluating the oral conditions. Consequently, the number of incoming patients would increase every year, resulting in unsustainable situations due to an insufficient workforce in dental and oral surgery clinics. If all surgical patients are automatically sent for oral management, the visits of those who do not require treatment will increase the burden on both patients and government finances. Lastly, oral management for all surgical patients may force medical providers to sacrifice the time for thoughtful explanation to patients, which might compromise the quality of care under the limited amount of preoperative time and human resources in the dental and oral surgery clinics. Thus, a reasonable solution that can affect sustainability is required.

We established the “oral triage” system in our hospital, wherein dental hygienists conduct an oral screening to select patients who need preoperative oral hygiene management based on the guidelines established by oral surgeons. For instance, they instruct patients with the risk of dislodging loose teeth during endotracheal intubation and extubation to visit a dentist or an oral surgeon, within the limitations of the dental hygiene instructions in the Dental Hygienists Act.

Among the several postoperative complications, such as wound infection and hemorrhage, the present study focused on postoperative pneumonia because it is known to be affected by poor oral hygiene [1]. The aim of this study was to statistically examine the effectiveness of preoperative oral management, using a developmental “oral triage” system, in preventing postoperative pneumonia and the sustainability of the system on surgical patients with cancer who were initially introduced into the Japanese universal health insurance system. In addition, we present a comparative analysis of our findings with those of other studies that evaluated the conventional method.

### 1.1. Center for Perioperative Medicine and Oral Management

In April 2011, the Center for Perioperative Medicine was developed in the central surgery section to predict various perioperative complications under general anesthesia. The surgeon in charge is obligated to strictly comply with the rule to send patients to the Center for Perioperative Medicine at least two weeks before the surgery, except in case of an emergency.

The Center is comprised of anesthesiologists, surgical nurses, pharmacists, dental hygienists, clinical engineers, a Medical Affairs section, and a Properties Management section. Medical personnel, such as surgeons, nurses in outpatient clinics, and oral surgeons or dentists, collaborate with the members in the Center to constitute team care.

The patient flow at the Center for Perioperative Medicine is presented in Figure 1. To begin with, the surgeon in charge refers the patient to the Center for Perioperative Medicine via the outpatient nurse. At the center, a general explanation about the surgical procedure and perioperative management is provided by the surgical nurses using iPad movies. This is followed by the provision of instructions regarding drug compliance and discontinuation of medications prior to surgery by a pharmacist. Oral screening is performed, and instructions are provided by a dental hygienist, followed by the provision of an explanation about the methods used to prevent deep vein thrombosis by a clinical engineer. Subsequently, an examination is conducted by an anesthesiologist using the complete information available through the electronic medical record system. This information includes the need for discontinuation of an antithrombotic drug, protection of mobile teeth at an oral surgery clinic, and application of a foot pump due to the high risk of deep vein thrombosis. This approach is an innovative method that involves teamwork.

In the oral management section, a dentist develops a treatment plan at an oral surgery outpatient clinic after oral screening at the Center for Perioperative Medicine. Professional oral care such as scaling, removal of the tongue coating, provision of oral hygiene instructions, extraction of teeth that can act as reservoirs of infection, and delivery of protective devices for the teeth (hereinafter referred to as the “Tooth Guard” in our hospital) is provided in the oral management section.

The curved arrow in Figure 1 indicates the patient flow for oral management using the conventional methods.

Subsequent patient education protocols should include instructions for self-care during professional oral care close to the day of surgery, as well as for instructions for cleaning dentures, rinsing with povidone-iodine mouthwash, and practicing oral self-care even on the morning of surgery when no food is eaten.

### 1.2. The “Oral Triage” System

In this system, a dental hygienist evaluates the oral hygiene, the intraoral reservoirs of infection, and the risk of dislodging or damaging the teeth under the management of an anesthesiologist (Table 1). Oral screenings, such as tooth mobility measurements, are performed by trained dental hygienists and censored to ensure uniformity in assessment [2]. Subsequently, the hygienist provides explanations about the need to visit the oral surgery clinic, if required. In the oral screening phase, patients are not charged a fee for the dental examination (included in the anesthesiologist’s consultation fee) and are required to report to the anesthesiologist if they need dental management. Subsequently, the patients who require dental management visit the oral surgery clinic on the same day with a referral letter from the anesthesiologist. An appointed dentist at the oral surgery outpatient clinic conducts examinations, develops treatment plans, and schedules appointments for the procedures. The delivery of a Tooth Guard and removal of the reservoirs for infection, including extraction and dental treatments, are performed by a dentist or an oral surgeon. On the other hand, professional oral care such as scaling, removal of tongue coating, and provision of oral hygiene instructions is provided by a dental hygienist. The initial visit fee and perioperative oral management fee are charged at this time.

All dental hygienists are trained to perform a standard level of professional oral care at least one or two times a week. The oral treatments were carried out one to three times before the operation. The aim was to achieve a plaque-free oral cavity immediately before surgery.

Oral care and instructions for oral hygiene management are provided once the patients are stabilized after the surgical procedure conducted under general anesthesia. Furthermore, a referral letter is sent to a family dentist requesting continuous management before discharging the patient from the hospital (Figure 2).

## 2. Methodology

The subjects in this study comprised patients who underwent surgery under general anesthesia at our hospital between April 2010 and March 2019. The sustainability and effectiveness of the system for cancer surgery patients were evaluated within this population. The items used for the analyses were the annual trends of the oral management execution rate, the trends of the incidence of postoperative pneumonia, and factor analyses of the onset of pneumonia by multivariate analyses.

The analyses were limited to cancer surgery because of the initial introduction of the patients into the Japanese social healthcare system and the lack of perioperative oral management for cancer patients prior to the addition to the fee schedule, which simplified the statistical analyses.

Oral management has been routinely advocated for patients who underwent cardiovascular surgical procedures and organ transplantations to prevent surgical site infection. However, prior to implementing this system, the concept of oral management before cancer surgery was non-existent (execution rate, 0%).

This retrospective study used data from the Diagnosis Procedure Combination (DPC) database system in Japan, the details of which have been described in the literature previously [3]. The DPC data of hospitalized patients included all electronic records pertaining to clinical and medical care information. Of all patients admitted to the participating institutions, only cancer patients who underwent surgery were selected using the DPC database.

Approval for the study was obtained from the ethics committee of the Toho University Oomori Medical Center (M16057).

### 2.1. Variables

The variables retrieved from the DPC database were as follows: sex, age, date of hospitalization, three categories of the names of the diseases (encompassing diseases that led to hospitalization, comorbidities at the time of hospitalization, diseases developed during hospitalization), and DPC codes.

### 2.2. Patient Selection

A total of 37,557 patients who underwent oral screening and surgery from April 2010 to March 2019 at our hospital were observed in this study. Subsequently, 7715 cancer surgery patients, excluding surgical patients with the onset of pneumonia at the time of the acceptance, were selected from this population to form the target group (Figure 3).

The annual trend in the rate of oral management execution was defined as the oral management execution rate per executed oral screening. The rate was at 0% for two years before the implementation of the Center for Preoperative Medicine. The data were collected for nine years from 2012 to 2018 after implementing the Center for Preoperative Medicine.

The first year between April 2012 and March 2013 was defined as the trial period, and the incidence rates of postoperative pneumonia two years prior to the introduction (pre-introduction; from April 2010 to March 2012) and the following eight years after the introduction (post-introduction) were compared. The following six years of post-introduction were divided into an early period (April 2013 to March 2015), middle period (April 2015 to March 2017), and late period (April 2017 to March 2019). With regard to the factor analyses of cancer type comparisons and the incidence of postoperative pneumonia, 1186 patients within the two years of pre-introduction and 2131 patients with the two years in the late period of post-introduction were compared (Figure 3).

### 2.3. Data Collection

The diseases at hospitalization in DPC were classified into “primary diseases for hospitalization”, “comorbidities at hospitalization”, and “diseases after hospitalization”. For example, when a patient with hypertension was hospitalized for the treatment of gastric cancer, the primary disease for hospitalization and the comorbidity at hospitalization were recorded as gastric cancer and hypertension, respectively, at the time of admission. If the patient experienced pneumonia after admission, the disease was additionally recorded under the section “diseases after hospitalization”.

Therefore, in the present report, patients with a record of any type of pneumonia in the “diseases after hospitalization” section who had no record of pneumonia in the “primary diseases for hospitalization” or “comorbidities at hospitalization” sections are considered as those with postoperative pneumonia. In other words, the disease is recorded only in the “diseases after hospitalization” section in patients who develop postoperative pneumonia, whereas patients without postoperative pneumonia would have no record of pneumonia in any of the three sections. Patients who had a record of any type of pneumonia in the “primary diseases for hospitalization” or “comorbidities at hospitalization” sections were considered as “excluded cases”.

The DPC database includes a 14-digit code called the DPC code. This code is specific to each patient and describes the cause of hospitalization, whether surgery or treatment was performed, the presence of side effects, and other details. The first six digits of the DPC code indicate the name of the disease. The DPC codes were used to classify the disease distribution of the sample population into 19 disease groups, and the target group was classified into 28 types of cancers. In the target group, the patients were classified based on the status of surgery; all cancer patients who underwent surgery were extracted from the database [3].

### 2.4. Statistical Analysis

Statistical analyses were performed using EZR ver1.40 (Kanda, 2014, Saitama, Japan) [4].

For a general understanding of the factors contributing to the development of pneumonia, age, sex, the incidence of postoperative pneumonia, and category of disease (cancer) were compared before (hereinafter referred to as pre-introduction) and after (hereinafter referred to as post-introduction) the introduction of the “oral triage” system in the sample population and the target group. Age was compared using a *t*-test, and sex and the incidence of postoperative pneumonia were compared using a Chi-square test. In the target group, the incidence of postoperative pneumonia by age and cancer type were calculated, and the incidence of pre-introduction was compared to that of post-introduction using the Chi-square test.

Based on the result, a multivariable logistic regression analysis was performed. The objective variable for the analysis was the onset of postoperative pneumonia, and the explanatory variables were pre-introduction and post-introduction of ‘oral triage’ system, sex, age, and cancer type. Age was classified into five variables (<50 years, 50–59 years, 60–69 years, 70–79 years, and ≥80 years), and cancer type was classified into five variables (brain, stomach, esophagus, lung, and others). A two-tailed *p*-value of <0.05 was considered statistically significant.

## 3. Results

### 3.1. Characteristics of the Sample Population and Target Group

The characteristics of the sample population and target group (pre-introduction and late period of post-introduction) are summarized in Table 2 and Table 3, respectively. Significant differences in age and sex were observed pre- and post-introduction in the sample population. However, no significant difference in the incidence of pneumonia was noted before and after the introduction. Alternatively, significant differences in age, sex, and incidence of pneumonia were observed pre-introduction and post-introduction in the target group. Diseases of the digestive system were found to be most common pre- and post-introduction in the sample group, with the number of patients increasing in the order of cardiovascular, female genitalia, puerperal, respiratory, and nervous system diseases. The most common disease in the target group was stomach cancer, followed by cancers of the colon, rectum and anus, liver and intrahepatic bile duct, and lung.

### 3.2. Trend in the Oral Management Execution Rate

The trend in the oral management execution rate after the introduction of the oral triage system from 2012 to 2018 is presented in Figure 4. The execution rate indicates the proportion of patients scheduled for surgery with oral problems, such as poor oral hygiene. The rates were found to range from 16.4% to 26.5%.

### 3.3. Trend in the Incidence Rate of Postoperative Pneumonia

The trend in the incidence rate of postoperative pneumonia in the target group was observed before and after implementing the “oral triage” system (Figure 5). The number of cancer patients showed an increasing trend over the years. However, the incidence rate of postoperative pneumonia decreased after the implementation of the oral management section from 2.07% before the introduction of the Center for Perioperative Medicine to 0.97% in 2018.

A significant difference in the incidence of postoperative pneumonia was observed between 2010 (pre-introduction) and 2018 (post-introduction; *p* = 0.04).

### 3.4. The Change in the Incidence of Postoperative Pneumonia Based on Cancer Type before and after Introduction of the “Oral Triage” System

The highest incidence rate of postoperative pneumonia was associated with surgery in patients with bone cancer, followed by those with esophagus, stomach, brain, lung, colon, uterus, and breast cancers before the introduction of the “oral triage” system. During the late period of post-introduction, the highest rate was associated with surgery in patients with skin cancer, followed by those with brain, esophagus, gallbladder and extrahepatic bile duct, lung, colon, prostate gland, and stomach cancers.

After the introduction of this system, the incidence of postoperative pneumonia was decreased in patients with cancers of the digestive organs, such as the stomach, esophagus, and colon, and of the lung, brain, bone, and uterus (Table 4). The introduction of this system significantly reduced postoperative pneumonia only in patients who underwent surgery for gastric cancer.

### 3.5. Factor Analysis for the Onset of Postoperative Pneumonia

The relative factors were examined using logistic regression analyses with the onset of postoperative pneumonia as the objective variable and pre- and post-introduction of the “oral triage” system, sex, age, and type of malignant tumors as the explanatory variables (Table 5).

When comparing the risk of postoperative pneumonia before and after the introduction of the “oral triage” system with corrections for sex, age, and type of malignant tumor, the odds ratio was significantly lower in the post-introduction group (late period) compared to that in the pre-introduction group (aOR = 0.50, *p* = 0.03).

The risk in males was about 4.5 times as high as that in females (*p* < 0.01). The odds ratio increased with an increase in age, demonstrating that patients ≥80 years (aOR = 12.90, *p* = 0.02) were at a significantly higher risk of postoperative pneumonia relative to those <50 years of age.

Furthermore, the odds ratio was found to be the highest in patients with brain tumors (aOR = 5.39, *p* < 0.01), followed by those with malignant esophageal tumors (aOR = 3.78, *p* = 0.02). The odds ratios in patients with malignant stomach and lung tumors were 2.21 and 2.03, respectively, but there was no statistically significant difference (Table 5).

## 4. Discussion

### 4.1. Introduction of the Concept of Perioperative Oral Management to the Universal Health Insurance System in Japan

Since April 2012, possible sources of dental infection such as caries and periodontal diseases were preoperatively treated to prevent the development of postoperative complications under the insured medical treatment system based on the universal health insurance system in Japan. However, due to the absence of dental-related complaints, oral hygiene and functional management of dental issues prior to surgery were not routinely conducted in patients with malignant tumors before 2012. Nevertheless, in 1999, Yoneyama et al. reported the preventive effects of oral care on aspiration pneumonia [5]. Likewise, in 2009, Akutsu et al. showed the preventive effects of oral care on the postoperative complications of esophageal cancer surgery [6]. These studies revealed the mechanism involved in the development of aspiration pneumonia as a consequence of disuse syndrome following extended cancer surgery and long-term bed rest. Furthermore, the possibility of preventing these types of pneumonia by oral care was suggested by dentists and oral surgeons. Based on these findings, the Ministry of Health, Labour, and Welfare listed “perioperative oral management” in the fee schedule for insured treatments in the social insurance scheme in April 2012. Although it is limited to organ transplantation, cardiovascular surgery, and all cancer surgeries under general anesthesia, oral management and treatments such as the protection of mobile teeth and removal of dental-related infectious reservoirs can be conducted under the perioperative oral management, despite the absence of dental-related complaints from the patients. The application was recently expanded to include patients with cerebrovascular and joint replacement surgery.

### 4.2. The Need for Analyses Using the “Oral Triage” System

A previous study reported that unifying the system, which included the scheduling of oral management and the techniques used for oral care in a multicenter study, is an issue that should be addressed [7,8,9,10,11]. Therefore, they conducted a statistical analysis using a large sample size by increasing the number of participating institutions to minimize the bias. In the system, which included data up to March 2019, the sample size per institution was approximately 37,000 when the statistical analyses were conducted using the “oral triage” system with unified techniques. The design used in the present study is unprecedented.

The present study did not make comparisons between interventional and non-interventional oral management; instead, it evaluated the differences before and after the introduction of the “oral triage” system. These comparisons aided in investigating the change in the incidence of postoperative pneumonia as a whole; thus, it is not necessary to consider the selection bias that accompanies whether oral management was performed or not.

In the “oral triage” system, dental hygienists were sufficiently trained to conduct oral screening evaluations in a standardized manner and select patients who needed to visit a dental or oral surgery clinic. The standardization, in terms of bacteriological and statistical credibility, and the uniformity in the evaluations have been evaluated and reported in a previous study and supported by the Japan Society for the Promotion of Science, Grant-in-Aid for challenging Exploratory Research (2015–2017) [2].

### 4.3. Sustainability of the “Oral Triage” System

In general, it is most common for a surgeon in charge to recommend patients to visit a dental or oral surgery clinic in the hospital. When a referral is made solely by a surgeon, patients with poor oral hygiene may not visit the dental clinic, and in some instances, the oral management might not be started before the surgery due to a last-minute referral. Moreover, as the in-house referral to a dental professional gains popularity, the number of incoming patients will increase every year, resulting in an unsustainable condition due to an insufficient workforce in the dental and oral surgery clinics. Oral management at the perioperative phase is becoming a standard procedure; yet, few hospitals have a sustainable system in place.

The patients are required to visit the Center for Perioperative Medicine two weeks prior to the surgery; consequently, the oral complications can be systematically resolved before the procedure. The rate of visits to our oral surgery clinics was around 20%. Considering the advantages with regard to the minimal human resources of doctors and dental hygienists performing oral management and treatments, our approach was considered sustainable.

Additional studies are required to investigate regional differences in the triage rates.

### 4.4. Effectiveness of ‘Oral Triage’ System in Preventing Postoperative Pneumonia in Cancer Patients

Several studies have reported the effectiveness of perioperative oral management in recent years. Nisio et al. conducted a retrospective study examining oral care intervention and the onset of postoperative complications in 27 intervention and 25 non-intervention cases among lung cancer patients who underwent thoracoscopic lobectomy [12]. The number of patients who had a body temperature of 38 °C or higher after surgery was significantly smaller in the intervention group, demonstrating a significantly shorter duration of hospitalization compared to the non-intervention patients.

Likewise, in the present study, the incidence rate of postoperative pneumonia in approximately 37,000 subjects was 2.07% before the introduction of the system and 0.97% six years after its introduction. The annual trends showed a gradual decrease in the incidence rates. A significant decrease was observed in 2018 (post-introduction). The possible reason for the gradual decrease might be an increase in the number of surgeons who referred patients to the Center for Perioperative Medicine strictly on a two-week deadline. Thus, if all surgeons adhere to this promise, the incidence of postoperative pneumonia is expected to decrease further.

In addition, among the patients who were triaged due to the need for oral management, some may have received multiple surgical treatments with repetitive explanations regarding the importance of oral management, which might have led to considerable improvements in preoperative oral hygiene, resulting in a decrease in the incidence of postoperative pneumonia over time.

### 4.5. Comparison of the Incidence of Postoperative Pneumonia Based on Cancer Type after Introduction of Oral Management System

The incidence of postoperative pneumonia was decreased in patients with malignant tumors of the stomach, esophagus, lung, colon, and brain after the introduction of the “oral triage” system in the present study. The fact that only patients with gastric cancer demonstrated a significant difference in the incidence of pneumonia revealed that the system might be effective for postoperative pneumonia after cancer surgery of the digestive system. Previously, Nishino et al. [12] compared the incidence rates of postoperative pneumonia and the duration of hospitalization in 50 intervention and 50 non-intervention patients with primary lung cancer, and 30 intervention and 70 non-intervention patients with esophageal cancer after the introduction of perioperative oral management. The incidence rate and duration of hospitalization were significantly improved in the lung cancer intervention patients but not in the esophageal cancer patients. Sotome et al. conducted a multicenter retrospective study on the risk factors of postoperative pneumonia in 234 intervention and 149 non-intervention esophageal cancer patients [11]. They found that the execution of oral function management (aOR = 0.42) was significantly associated with the onset of postoperative pneumonia. Therefore, to date, perioperative oral management is considered to be effective for the prevention of (postoperative) pneumonia after hospitalization in patients with malignant tumors of the stomach, esophagus, lung, and brain.

In the current study, subgroup analyses in each malignancy were not performed due to the possibility of obtaining inaccurate results owing to sample size reduction caused by group splitting. In the future, we aim to investigate the difference in the rate of oral management intervention due to the “oral triage” system in each malignant tumor.

### 4.6. Risk Factors for Postoperative Pneumonia

According to the results of the multivariable logistic regression analysis for the risk factors of postoperative pneumonia, the odds ratio after the introduction of the “oral triage” system was 0.50 (*p* = 0.03) times that before the introduction, thus indicating that the introduction of this system significantly reduced the incidence of postoperative pneumonia.

The odds ratio was higher in patients who were ≥50 years of age; hence, age was considered to be a relative factor. Teramoto et al. reported that 70% of all pneumonia patients who were ≥70 years presented with aspiration pneumonia [13]. Therefore, dysphagia associated with aging is more relevant as a risk factor of postoperative pneumonia than age itself. In the present study, the odds ratio for the onset of pneumonia in patients who were ≥80 years was 12.9 times higher (*p* = 0.02) than that in patients <80 years of age.

The risk of pneumonia among males was about four times as high as that in females (*p* < 0.01). The mortality rate of males due to pneumonia is higher than that of females in elderly people in Japan [14].

The risk of pneumonia was higher in patients with brain tumors (aOR = 5.39, *p* < 0.01) than in those with esophageal cancer (aOR = 3.78, *p* = 0.02). Furthermore, it was suggested that impaired consciousness and dysphagia commonly occurred as sequelae of craniotomy for brain tumors. In addition, esophageal cancer patients who underwent surgery were presumed to have postoperative pneumonia due to gastroesophageal reflux; it was concluded that patients with esophageal cancer have the second-highest incidence of postoperative pneumonia after brain tumors.

Taken together, these findings demonstrate the importance of preoperative evaluation for dysphagia and perioperative oral management in patients undergoing surgery for brain tumors and esophageal cancer in order to decrease the incidence of future postoperative pneumonia to nearly 0%. Furthermore, it will be important to predict the postoperative prognosis of dysphagia, recover the function immediately after the onset of dysphagia, and ensure oral management.

In the future, we plan to verify the incidence of postoperative pneumonia between the oral care intervention group and the non-intervention group using the same database but limited to patients undergoing surgery for a brain tumor and esophageal cancer.

### 4.7. Other Factors for Preventing Postoperative Pneumonia

A reduction in the onset of postoperative pneumonia in cancer patients was observed after the establishment of the oral management section at the Center for Perioperative Medicine and the listing of a perioperative oral management fee in the social insurance scheme since 2012. However, it cannot be proven that these are the only factors responsible for the reduction. In addition to performing oral management, the oral surgeons and dental hygienists in our department participate in the activities of the Swallowing Dysfunction Management Team and the Respiration Support Team for the prevention of ventilator-associated pneumonia. They seamlessly manage the oral cavities of the hospitalized surgery patients and contribute to the decrease in postoperative pneumonia. Moreover, the oral surgery department hosts oral hygiene management workshops to educate nurses and improve their awareness and skills. This widespread understanding of oral management might have had positive effects in reducing the incidence of postoperative pneumonia.

### 4.8. Comparison between the Present System and a Similar Study

Kurasawa et al. published a new topic on the effect of the implementation of oral management into the social insurance scheme to prevent the onset of postoperative pneumonia [15] (Table 6). They collected data from eight institutions that used the conventional method without defining the method and period (Division 5 in Table 7). The dentists or oral surgeons evaluated the incidence of postoperative pneumonia in the surgery patients. One of the limitations of their study was that they had obtained the results without providing any details about the oral care or the percentage of dental intervention provided to the preoperative patients. However, the need for a prospective study with standardized procedures using specific guidelines was mentioned. In the present study, the rates of oral management intervention were calculated, and the incidence rates of postoperative pneumonia were found to gradually decrease with systemization. Furthermore, oral care treatments were provided in a standardized manner by trained staff within the same department.

The incidence rate of pneumonia during hospitalization as a result of introducing perioperative oral management to the insurance system was reported to be 0.8% in the study by Kurasawa et al. [15]. If the localized effect on the incidence rate is taken into consideration, this result suggests that the conventional method seems to be sufficiently effective. However, issues such as insufficient time for oral management and inconsistent daily referral numbers may result in unmanageable appointments on a busy day. To obtain stable and sustainable results, we consider the “oral triage system” to be advantageous. This system mandates the start of oral management 14 days before surgery; hence, one dentist and one hygienist can manage the cases even if the number of surgery cases increases. Although the incidence rate of pneumonia in the report by Kurasawa et al. was lower than that in the current study (0.97%), but we believe that the introduction of the “oral triage” system has several advantages.

Ishimaru et al. conducted a study on the preventive effects of oral care on the onset of pneumonia among patients after malignant tumor surgery using the National Database of Health Insurance Claims and Specific Health Checkups of Japan (NDB) analysis [16,17] (Table 6). The NDB is a database developed by the Ministry of Health, Labour and Welfare, which collects information on health insurance claims and specific health checkups/specific health guidance from 2009 under the “Act on Assurance of Medical Care for Elderly People”. The NDB data has been provided to third parties, particularly researchers, since 2011.

In an observational study using the database, the intervention group is not randomly allocated when the effects of the oral intervention group are compared to that of the non-intervention group. Thus, the backgrounds of the patients vary, and patient selection bias may affect the estimated results. Therefore, a method using the propensity scores (PS) has been used, wherein background items, including comorbidities affecting the intervention effects, are included in the model as covariates to adjust the effects of the intervention group.

High-risk cases of postoperative pneumonia were present among the non-intervention subjects in the 2018 study by Ishimaru et al. [16] due to poor oral hygiene before surgery; nonetheless, many patients having surgery with good oral hygiene were also included. Therefore, it must be considered that the analysis was conducted with unknown proportions of patients with good oral hygiene, which could have an effect in lowering the incidence rates of postoperative pneumonia. On the contrary, although several appointments are required to improve oral hygiene, the oral care intervention group included patients with poor oral hygiene who underwent surgery without sufficient treatments due to the limited number of days until surgery. Hence, the effects of these factors on the results must be considered. Although the adjustment using the disease periodontitis was conducted between the two groups, it does not sufficiently reflect the oral hygiene conditions. Therefore, the background factors between the two groups in the study by Ishimaru et al. were not adjusted in terms of oral hygiene immediately before surgery [16] and might have had a substantial impact on the results of the statistical analysis. In addition, from the viewpoint of the oral management classifications in Japan (Table 7), it must be acknowledged that there is a mixture of all methods of oral management in the clinical visiting systems of NDB. On the other hand, the “oral triage system” in the current study falls under one category in the oral management system classification. This system (Division 8 in Table 7) ensures enough days for the completion of treatments in patients with poor oral hygiene before surgery.

Ishimaru et al. reported that the oral intervention rates in all patients and the incidence rate of pneumonia in patients with oral intervention were 16% and 3.28%, respectively [16]. This incidence rate can be further decreased if the oral intervention rate is improved by evaluating the preoperative oral hygiene conditions. Thus, the present study compared the groups of patients two years pre-introduction (2010–2011) and two years post-introduction (2017–2018). There was no selection bias because the treatment interventions were not compared. Therefore, in addition to displaying the collected results with the background factors in the two groups, multivariate analyses were conducted without adjusting for the background factors. As a result, the incidence of postoperative pneumonia had decreased to 0.97% with oral intervention rates of 20%, thereby confirming that this new system was significantly effective.

## 5. Future Study

Although postoperative aspiration pneumonia was the main focus of this study, additional studies are required to investigate whether the “oral triage” strategy can be used to prevent conditions such as wound infection caused by poor oral hygiene from the perspective of periodontal medicine [18].

## 6. Conclusions

Using the “oral triage” system, we were able to successfully remove the reservoirs of oral infection within a sufficient treatment time frame of two weeks. In addition, the incidence rate of pneumonia was lowered by employing limited human resources to render oral management relative to the conventional method, thereby emphasizing the superiority of this system. The system proved to be a sustainable and developmental perioperative oral management strategy.

## Figures and Tables

**Figure 1 ijerph-18-06296-f001:**
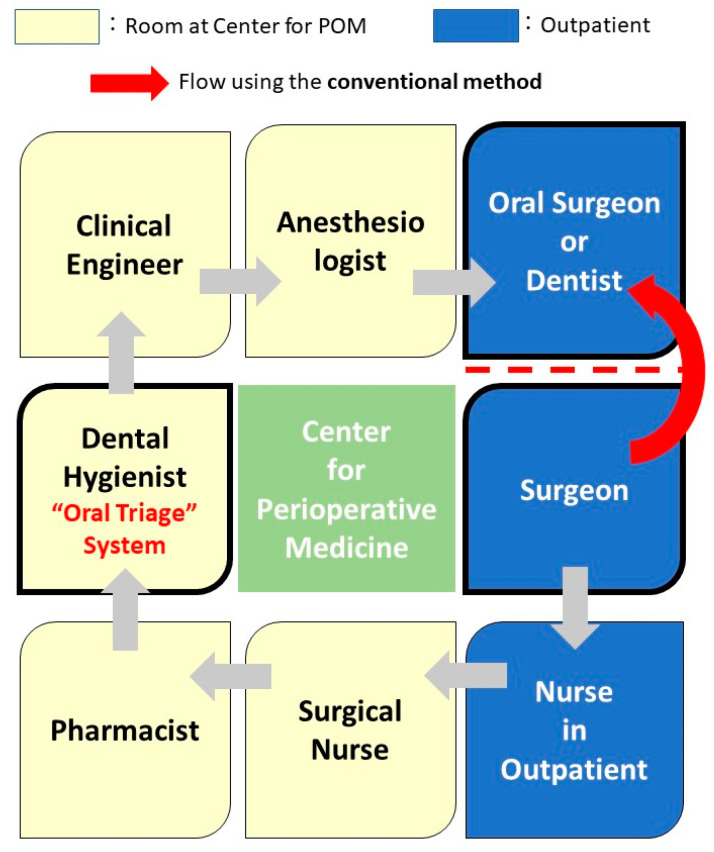
Patient flow at the Center for Perioperative Medicine and “oral triage”. In the “oral triage” system, the surgeon is the starting point. The patient is referred to various departments (as shown in a clockwise direction in this schematic figure), and finally, the oral management is performed by the oral surgeon. In the conventional method, as shown by the curved red arrow, the surgeons directly consult the oral surgery clinic.

**Figure 2 ijerph-18-06296-f002:**
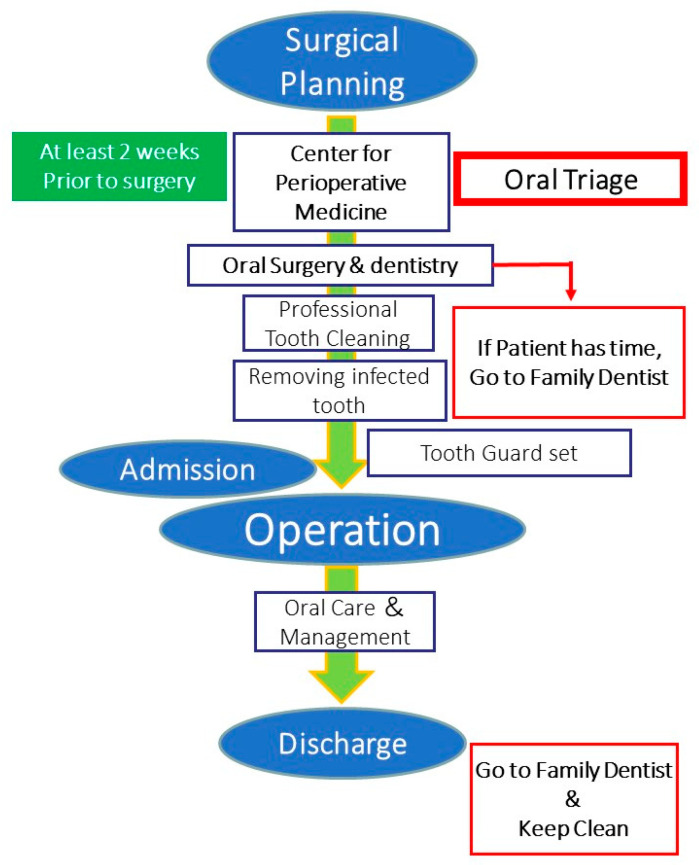
Unique “oral triage” system for Perioperative Oral Management. The patient is required to visit the perioperative center at least two weeks prior to the surgical procedure to assess and evaluate the oral and medical conditions (including the drug history) in order to avoid any complications during general anesthesia.

**Figure 3 ijerph-18-06296-f003:**
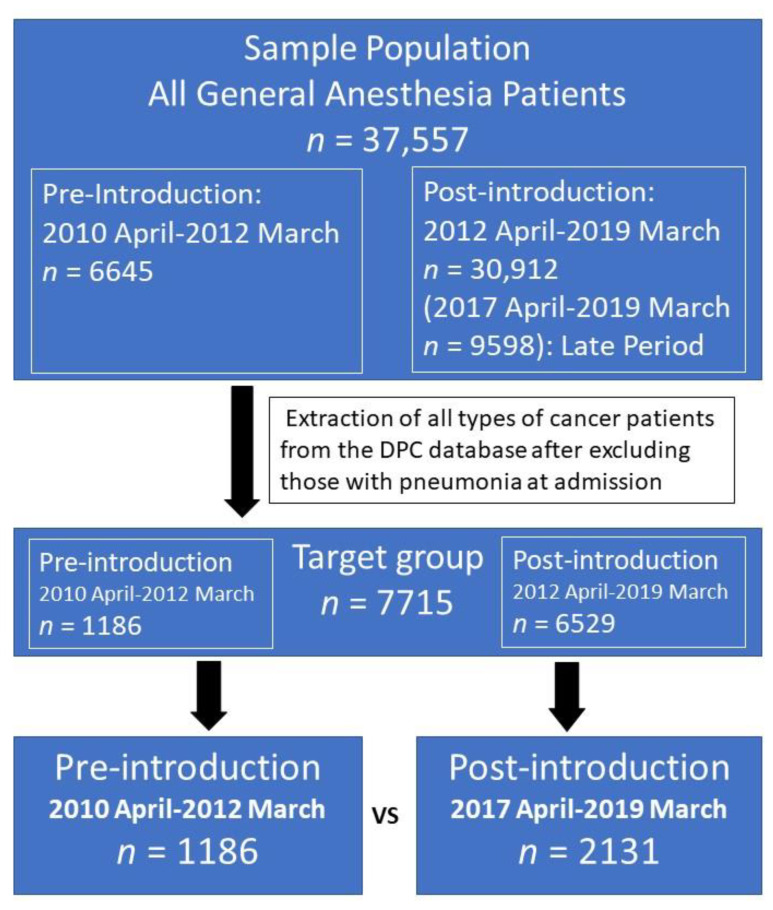
Schematic representation of patient selection. The final comparison is between the target patient groups two years prior to the introduction of the “oral triage” system and the last two years after its introduction.

**Figure 4 ijerph-18-06296-f004:**
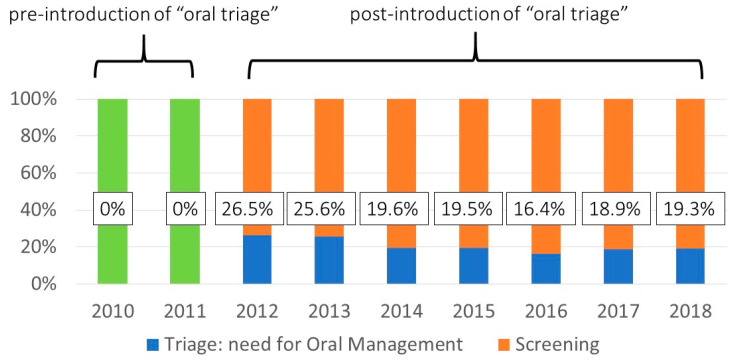
Changes in the number of ‘triage’ patients. The graph shows the yearly trend in the percentage of patients who underwent oral screening and required oral management. The percentage was ‘zero’ before the introduction of the “oral triage” system. The green bars show all surgical patients when no screening was performed.

**Figure 5 ijerph-18-06296-f005:**
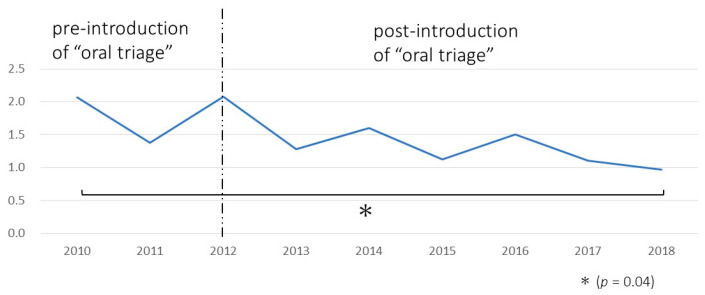
Changes in the incidence rate of postoperative pneumonia in the target group. The incidence rate was significantly reduced in 2018, after the introduction.

**Table 1 ijerph-18-06296-t001:** Screening of oral condition. If either items two or three in these check criteria apply, the patient should be instructed to go to an oral surgery clinic in our hospital, and the details should be reported to the anesthesiologist.

**Teeth Movement (0–3)**
0. Normal
1. Slight
2. Horizontal 4 direction
3. 2+ Vertical
**Gingiva Inflammation (0–3)**
0. Normal
1. Redness without bleeding
2. Redness, edematous with bleeding
3. Redness, edematous with spontaneous bleeding or bleeding on pressure
**Dental Calculus (0–3)**
0. None
1. Within one-third of the dental surface
2. One-third to two-thirds of the dental surface or spot calculus in the gingival sulcus
3. More than two-thirds of the dental surface or band calculus in the gingival sulcus
**Dental Plaque (0–3)**
0. None
1. Within one-third of the dental surface or attached to a foreign body
2. One-third–two-thirds of the dental surface
3. More than two-thirds of the dental surface
**Tongue (1–3)**
1. Pink, moist, and presence of papillae
2. Loss of papillae with redness
3. Heavy tongue coating with or without ulceration
**Xerostomia (0–3)**
0. Normal
1. Sticky tongue
2. Foamy saliva
3. Cracks on the tongue
**Halitosis (1–3)**
1. None
2. Recognized within a distance of 30 cm
3. Recognized over a distance of 30 cm
**Dysphagia (1–3)**
1. Normal
2. Difficulty swallowing
3. Unable to swallow
**Trismus (1–3)**
1. Normal
2. Trismus, but able to open the mouth on their own
3. Trismus as one finger width

**Table 2 ijerph-18-06296-t002:** Characteristics of the sample population.

		2010–2011 (*n* = 6645)		2017–2018 (*n* = 9598)		*p*-Value
Sex, *n* (%)	Male	3018	(45.4)	4933	(51.4)	-
	Female	3627	(54.6)	4665	(48.6)	0.0000645
Age	Ave. ± SD	49.68 ± 22.72	-	52.32 ± 22.65	-	0.000000
	Mean	52	-	56	-	-
Hospitalization days	Ave. ± SD	14.57 ± 24.03	-	16.55 ± 24.41	-	0.000000
	Mean	8	-	9	-	-
Post-operative pneumonia, *n* (%)		56	(0.8)	86	(0.8)	0.73300
Type of disease, *n* (%)	Nervous system	106	(1.6)	221	(2.3)	-
	Ophthalmic	121	(1.8)	195	(2.0)	-
	Otolaryngological	411	(6.2)	1031	(10.7)	-
	Respiratory	175	(2.6)	390	(4.1)	-
	Cardiovascular	104	(1.6)	356	(3.7)	-
	Digestive system	1231	(18.5)	1814	(18.9)	-
	Musculoskeletal	823	(12.4)	1273	(13.3)	-
	Subcutaneous	65	(1.0)	61	(0.6)	-
	Breast	239	(3.6)	311	(3.2)	-
	Endocrine and Metabolic	121	(1.8)	173	(1.8)	-
	Renal urinary tract and male genital	1122	(16.9)	1259	(13.1)	-
	Female genitalia and puerperal	1466	(22.1)	1620	(16.9)	-
	Blood hematopoiesis	34	(0.5)	53	(0.6)	-
	Neonatal	306	(4.6)	333	(3.5)	-
	Trauma, burn, Poisoning	0	(0.0)	3	(0.0)	-
	Pediatrics	223	(3.4)	305	(3.2)	-
	Mental illness	0	(0.0)	5	(0.1)	-
	Others	98	(1.5)	195	(2.0)	-

*n*: number of patients; Ave.: average; SD: standard deviation.

**Table 3 ijerph-18-06296-t003:** Characteristics of the target group.

		2010–2011 (*n* = 1186)		2017–2018 (*n* = 2131)		*p*-Value
Sex *n* (%)	Female	685	(57.8)	1090	(51.1)	-
	Male	501	(42.2)	1041	(48.9)	0.00028
Age	Ave. ± SD	61.28 ± 15.16	-	63.97 ± 14.41	-	0.000000442
	Mean	64	-	64	-	-
Hospitalization days	Ave. ± SD	22.19 ± 30.65	-	20.75 ± 23.03	-	0.127
	Mean	14	-	14	-	-
Post-operative pneumonia, *n* (%)		20	(1.69)	22	(1.03)	0.108
Cancer type, *n* (%)	Brain	42	(3.5)	75	(3.5)	-
	Cornea, eye, and Appendage	0	(0.0)	6	(0.3)	-
	Head and neck	8	(0.7)	41	(1.9)	-
	Mediastinal	7	(0.6)	15	(0.7)	-
	Lung	92	(7.8)	232	(10.9)	-
	Esophageal	28	(2.4)	62	(2.9)	-
	Stomach	102	(8.6)	175	(8.2)	-
	Small intestine and Peritoneum	5	(0.4)	24	(1.1)	-
	Colon	236	(19.9)	406	(19.0)	-
	Liver and intrahepatic bile Duct	32	(2.7)	78	(3.7)	-
	Gallbladder and extrahepatic bile Duct	11	(0.9)	34	(1.6)	-
	Pancreas	17	(1.4)	53	(2.5)	-
	Bone	7	(0.6)	4	(0.2)	-
	Soft tissue	18	(1.5)	32	(1.5)	-
	Melanoma	9	(0.8)	10	(0.5)	-
	Non-melanoma skin	19	(1.6)	17	(0.8)	-
	Brest	222	(18.7)	283	(13.3)	-
	Thyroid gland	44	(3.7)	71	(3.3)	-
	Renal	40	(3.4)	81	(3.8)	-
	Genital	1	(0.1)	3	(0.1)	-
	Renal pelvis and Ureter	17	(1.4)	35	(1.6)	-
	Prostate	40	(3.4)	136	(6.4)	-
	Ovary and uterine appendage	32	(2.7)	67	(3.1)	-
	Cervix and uterine Body	131	(11.0)	148	(6.9)	-
	Vulva	2	(0.2)	2	(0.1)	-
	Hematological malignancy	17	(1.5)	35	(1.6)	-
	Other	7	(0.5)	6	(0.4)	-

*n*: number of patients; Ave.: average; SD: standard deviation.

**Table 4 ijerph-18-06296-t004:** Comparison of the incidence of pneumonia based on cancer type. The incidence of postoperative pneumonia was decreased in patients with cancers of the digestive organs, such as the stomach, esophagus, and colon, and of the lung, brain, bone, and uterus, as a result, but only in gastric cancer surgery had a significant difference in decrease.

	2010 + 2011 (*n* = 1186)			2017 + 2018 (*n* = 2131)		
Cancer Type	Patients	Pneumonia	Rate	Patients	Pneumonia	Rate
Brain	42	2	4.76	75	3	4.00
Cornea, eye, and appendage	0	0	0	6	0	0
Head and neck	8	0	0	41	0	0
Mediastinal	7	0	0	15	0	0
Lung	92	2	2.17	232	4	1.72
Esophageal	28	2	7.14	62	2	3.23
* Stomach	102	7	6.86	175	1	0.57
Small intestine and Peritoneum	5	0	0	24	0	0
Colon	236	4	1.69	406	5	1.23
Liver and intrahepatic bile Duct	32	0	0	78	0	0
Gallbladder and extrahepatic bile Duct	11	0	0	34	1	2.94
Pancreas	17	0	0	53	0	0
Bone	7	1	14.29	4	0	0
Soft tissue	18	0	0	32	0	0
Melanoma	9	0	0	10	0	0
Non-melanoma skin	19	0	0	17	2	11.76
Brest	222	1	0.45	283	0	0
Thyroid gland	44	0	0	71	0	0
Renal	40	0	0	81	0	0
Genital	1	0	0	3	0	0
Renal pelvis and ureter	17	0	0	35	0	0
Prostate	40	0	0	136	1	0.74
Ovary and uterine appendage	32	0	0	67	0	0
Cervix and uterine body	131	1	0.76	148	0	0
Vulva	2	0	0	2	0	0
Hematological malignancy	17	0	0	35	3	33.33
Other	7	0	0	6	0	0

* *p* < 0.01.

**Table 5 ijerph-18-06296-t005:** Risk factors for postoperative pneumonia in the target group by logistic regression analysis.

	Number of Patients	Odds Ratio (95% CI)	*p*-Value
**Install “Oral Triage” System**			
Pre-introduction: 2010 April–2012 March	1186	standard	-
Post-introduction: 2012 April–2019 March	2131	0.50 (0.27–0.94)	**0.031**
**Age (years)**			
Younger than 50	609	standard	-
50–59	524	4.44 (0.51–38.60)	0.18
60–69	917	5.51 (0.71–43.00)	0.1
70–79	919	5.11 (0.65–40.30)	0.12
80 or older	348	12.90 (1.60–103.00)	**0.016**
**Sex**			
Female	1775	standard	-
Male	1542	4.51 (1.95–10.40)	**0.00042**
**Cancer type**			
Others	2509	standard	-
Brain	117	5.39 (1.93–15.10)	**0.0013**
Esophagus	90	3.78 (1.23–11.60)	**0.02**
Stomach	277	2.21 (0.94–5.21)	0.068
Lung	324	2.03 (0.80–5.16)	0.14

CI: confidence interval. Bold: *p*-values indicate significance.

**Table 6 ijerph-18-06296-t006:** Comparison with other similar studies. Pneumonia incidence rate after introducing oral care intervention is abbreviated to PIRO.

Author	Sample Size	Method	System (Execution Rate)	Years	Results (PIRO)	Method of Oral Management
Present study	7715	individual, raw DPC data retrospective, and MLR	oral triage (20%) Division 8	2010–2018: 9 years	significant (0.97%)	same
Kurasawa Y. et al., 2020 [15]	25,554	multicenter, raw DPC data, retrospective, and MLR	Conventional (unknown) Division 5	2010–2013: 4 years	significant (0.81%)	different between units
Ishimaru M. et al., 2018 [16]	509,179	NDB of Japan, retrospective, and IPTW	description of any dental treatment (16%)	2012–2015: 3 years, 7 months	significant (3.28%)	different between units

**Table 7 ijerph-18-06296-t007:** Classifications of the Japanese Oral Management system for surgery.

Type of Patient Who Visited the Oral Surgeon or Dentist	No Check Gate until General Anesthesia	Check Gate until General Anesthesia
**All Patients**	Division 1: Type of appointment for oral management	Division 2: Perioperative management center (ALL)
**From a specific section of surgery** **(ex. All patient from** **gastrointestinal surgery only)**	Division 3: Type of surgeons-dentists collaboration	Division 4: Perioperative management center (specific)
**Surgeon’s decision**	Division 5: Conventional method	Division 6: Not exist
**Oral surgeons or dentists** **decision, or after performing a screening test**	Division 7: System of “oral triage” by dental section	Division 8: This study System of “oral triage” by center

## Data Availability

The data presented in this study are available on request from the corresponding author. The data are not publicly available due to privacy and ethical considerations.

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
