# Peer review of "Preventive Effects of Sustainable and Developmental Perioperative Oral Management Using the “Oral Triage” System on Postoperative Pneumonia after Cancer Surgery"

_ijerph, 2021, doi:10.3390/ijerph18126296_

Round 1

Reviewer 1 Report

Dear Author,

Minor revisions are required. All of my comments are provided in the attached file in the form of comments/sticky notes. Kindly provide a reply/rebuttal to the comments and implement corresponding revisions in the revised manuscript. Based on the revised draft, I will provide further recommendations

Reviewer 2 Report

In general, this is an interesting and well-written manuscript. The results may provide the merits in this research.  

The concepts of the patient-centered comprehensive dental treatment and the team resource management are suggested to discuss in the text.

English editing is required.

After revision, this manuscript could be accepted for publication.

Author Response

Answer: Thank you for your appreciative and constructive comments. Patient-centered comprehensive dental treatment and team resource management have been added to the Discussion section as per your suggestion. This manuscript has been reviewed by a native English speaker attached with certificate.

Reviewer 3 Report

Dear Authors,

Congratulations on the successful completion and submission of your research manuscript for the peer-review process.

The research study introduces a sustainable and developmental perioperative oral management using the 'oral triage system' for cancer patients undergoing surgical procedures.

Unlike the routine system where these patients are scheduled for dental visits with a dentist/ oral surgeon prior to the surgery, the triage system introduces the strategy of streamlining these patients primarily to a trained dental hygienist who does the initial oral evaluation and thereafter, only the patients who require further treatment, so as to prepare them for the surgery are referred to dentists/oral surgeons for oral management. In this way, the system efficiently addresses concerns regarding - limited workforce available to render perioperative oral management; elimination of any potential compromise in the quality of oral care provided to these patients; and most importantly the reduced incidence rate of post-operative pneumonia in these patients.

Strengths of the study include: introduction of a sustainable system for efficient and effective oral perioperative management, clear description of the research methodology, well-presented results, adequately supported discussion with elaborate comparisons with similar studies in the literature, inclusions of future perspectives of the research study. 

The authors are advised to consider the following suggestions:

  • Under introduction, the authors could include a sentence or two about the rationale and significance behind studying the association between perioperative oral management and post-operative pneumonia.
  • Line 17: can change '...will increase...' to '...were seen to increase..'
  • Line 30: may consider to mention as ...'...developmental oral management strategy to...'
  • Line 58: change '..those visits...' to 'the visits..'
  • Line 67: instead of 'In addition,..' the authors may consider to write it as 'For instance, they...' (because that is one of the reasons why they dental hygienist would refer the patient to oral surgeon for further management and not the only reason as per Table 1)
  • Line 72: change to '...system on surgical patients..'
  • Line 87: can modify as 'To begin with, the surgeon in-charge...'
  • Line 87: '...to the Centre...'
  • Line 131: change '..include' to '..included..'
  • Line 132: can rephrase the sentence to something like this: 'Subsequently, these patients requiring dental management visit the oral surgery clinic on the same day with a referral letter from the anaesthesiologist'.
  • Line 134: change 'one dentist' to 'an appointed dentist'
  • Line 137: '...or an oral surgeon..'
  • Line 137: can change to 'On the other hand, professional oral care such as..'
  • Line 141: '... are trained to perform a standard..'
  • Line 142: based on what? Insurance system? Kindly complete the sentence
  • Line 145: can mention as '... provided once the patients stabilise after the surgical procedure under general anaesthesia'
  • Line 146: 'Furthermore, a referral letter..'
  • Line 150: 'Upto two weeks..' please rephrase this whole sentence to provide better clarity to the readers
  • Line 164: 'Oral management has been routinely advocated to patients..'
  • Line 169: please include a reference as indicated
  • Figure 3 - could be made to look more appealing
  • Line 214: change to 'diseases after hospitalization' section...(representing within quotes - wherever needed, throughout can make it easily absorbable for the readers).
  • Table 6: First column - can change to something like 'Types of patients who visited the oral surgeons or dentists' 
  • Line 393: '...which was supported by...'
  • Line 602: can modify to something like this: 'In addition, the incidence rate of pneumonia was lowered by employing limited human resources to render oral management relative to the conventional method, thereby highlighting/emphasizing the superiority of this system. The system indeed proved to be a sustainable and developmental perioperative oral management model/strategy'.

Good Luck!

Reviewer 4 Report

The topic is really interesting and the manuscript is well organized.

The study concerning the perioperative oral management is interesting and original, especially considering its demonstrated sustainability and applicability.

The submitted study may pave the way for newly introduced preventative perioperative measures, concerning other post-operative systemic infections, beyond pneumonia, and focusing on the multi-/inter-disciplinary approach to surgical and, especially, cancerous patients.

The manuscript is well organized and well-written, although a little long-winded, therefore, I would suggest to make it more concise, in order to be more readable.

Author Response

Answer: Thank you for your kind comments and suggestions. As you mentioned, the introduction and discussion sections were long and there was some overlap. They have been reduced in the revised manuscript for better readability.

We hope that it is acceptable for publication in its current form.

Reviewer 5 Report

The article is very interesting but some considerations should be taken into account to facilitate its reading

The aim

The objective must be concise, for example effectiveness of the protocol. “In addition, we present a comparative analysis of our findings with those of other studies that evaluated the conventional method”, this is part of a good discussion.

Introduction

Should be abbreviated

Methodology

The authors must present the protocol, in a structured way.

Regarding the "Screening of oral condition":

- What do you use to determine teeth movement?

- By what criteria do you analyze “Ulceration with fungal infection”?

- Don't consider tooth decay?

- Why do you consider blisters as a parameter of xerostomia?

What are the dental hygiene instructions given to patients? and, which one for the hygiene of the prosthesis?

Based on items 2 or 3 of the screening, what do you propose as treatment in two weeks prior to surgery?

What hygiene protocol do you recommend while the patient is admitted, such as Labeau et al., or another.

Results

What are the results of the multivariate study, what confounding factors have you considered?

Line 273: What parameters do you use to consider poor hygiene?

Discussion

It should be abbreviated and concise

References

Adapt them to the guide of the journal.

Round 2

Reviewer 5 Report

Thank you very much for your correction. With the best regards